# GDF-15 Levels in Gouty Arthritis and Correlations with Decreasing Renal Function: A Clinical Study

**DOI:** 10.3390/biomedicines13071767

**Published:** 2025-07-18

**Authors:** Osman Cure, Ertugrul Yigit, Merve Huner Yigit, Hakki Uzun

**Affiliations:** 1Department of Rheumatology, Faculty of Medicine, Recep Tayyip Erdogan University, Rize 53000, Turkey; 2Department of Medical Biochemistry, Faculty of Medicine, Karadeniz Technical University, Trabzon 61080, Turkey; ertugrulyigit@ktu.edu.tr; 3Department of Medical Biochemistry, Faculty of Medicine, Recep Tayyip Erdogan University, Rize 53000, Turkey; merve.huner@erdogan.edu.tr; 4Department of Urology, Faculty of Medicine, Recep Tayyip Erdogan University, Rize 53000, Turkey; hakki.uzun@erdogan.edu.tr

**Keywords:** GDF-15, gouty arthritis, renal function, biomarker, inflammation

## Abstract

**Background/Objectives**: Gouty arthritis (GA) is a chronic inflammatory disorder frequently linked to systemic inflammation and impaired kidney function. Growth differentiation factor-15 (GDF-15) has been suggested as a potential biomarker involved in both inflammatory responses and renal dysfunction. Studies on GDF-15 serum levels and renal function decline in GA patients are limited. This study aimed to investigate serum GDF-15 levels in patients with GA and to evaluate the relationship between GDF-15 and renal function parameters. **Methods**: This prospective case–control study included 60 (intercritical group: 30; acute attack group: 30) patients with gout arthritis and 60 healthy controls, matched for body mass index and sex. The enzyme-linked immunosorbent assay measured serum GDF-15, and renal function and inflammatory markers were also assessed. Group comparisons used non-parametric tests, Spearman’s analysis evaluated correlations, and receiver operating characteristic (ROC) analysis assessed diagnostic performance. **Results**: Serum GDF-15 levels were significantly higher in GA patients than controls (*p* < 0.001), especially during acute attacks. GDF-15 correlated moderately with renal function markers. ROC analysis showed high diagnostic accuracy for both acute (area under the curve (AUC) = 0.98) and intercritical gout phases (AUC = 0.96). **Conclusions**: Serum GDF-15 levels are increased in patients with gouty arthritis and are associated with impaired renal function. GDF-15 may serve as a helpful biomarker for disease activity and renal involvement in GA, but its interpretation should be considered in conjunction with other clinical and laboratory parameters.

## 1. Introduction

Gouty arthritis (GA) is an inflammatory joint disease that poses a significant health concern, particularly affecting men and postmenopausal women. Its global prevalence reaches up to 6%, although rates vary depending on geographic region and ethnicity [1,2]. In gouty arthritis, prolonged elevation of uric acid concentrations in the bloodstream results in the deposition of monosodium urate (MSU) crystals within synovial structures, various tissues, and organs. This crystal accumulation triggers acute inflammatory episodes, and if left unmanaged, can progress to chronic joint disease and the formation of tophi. The persistent inflammatory response may impair joint integrity and organ function, thereby contributing to higher rates of illness and death [3]. The treatment of gout typically involves colchicine, corticosteroids, non-steroidal anti-inflammatory drugs (NSAIDs), and urate-lowering therapies. A central component in the pathogenesis of gout is the activation of the inflammasome known as the NOD-like receptor family pyrin domain-containing 3 (NLRP3). This intracellular protein complex detects cellular danger signals and activates caspase-1, which subsequently cleaves pro-interleukin-1β into its active form, IL-1β, thereby initiating a strong inflammatory response. Diagnostic evaluation includes a clinical assessment, measurement of serum urate concentration, and assessment of inflammatory markers such as the C-reactive protein (CRP) and erythrocyte sedimentation rate (ESR), as well as proinflammatory cytokines including IL-1 and IL-6. Imaging modalities, such as ultrasonography and magnetic resonance imaging, can aid in diagnosis; however, the definitive diagnostic criterion is the identification of monosodium urate crystals in synovial fluid under polarized light microscopy [4,5,6]. These methods and markers are disadvantageous for diagnosis because they are not specific to the disease, and are expensive and invasive [7]. Therefore, there is a continuing need for specific, inexpensive, and easily applicable new biomarkers that can guide treatment and prevent attacks and complications of gouty arthritis.

Growth and differentiation factor fifteen (GDF-15), first described in 1997, is a signaling protein classified as a member of the transforming growth factor beta (TGF-β) family of cytokines [8]. GDF-15 is produced in a variety of tissues and organs, including the placenta, kidneys, lungs, heart, and macrophages. It is involved in immune regulation during both normal physiological states and pathological conditions, such as inflammation, tissue injury, and oxidative stress. It plays a key role in modulating the immune response to infectious and non-infectious diseases [9]. GDF-15 helps decrease inflammation by inhibiting proinflammatory molecules, including tumor necrosis factor-alpha, interleukin-1, and interleukin-6, functioning similarly to anti-inflammatory cytokines like interleukin-10, and thereby protecting organs from inflammatory damage. Beyond its beneficial role in conditions related to metabolism, such as diabetes and obesity, GDF-15 has also been suggested as a potential biomarker for assessing the severity and risk of death in illnesses like cardiovascular disease, malignancies, sepsis, and coronavirus disease 2019 [10,11]. One study demonstrated that individuals diagnosed with rheumatoid arthritis, a long-standing inflammatory condition, exhibited elevated concentrations of GDF-15 in their blood [12]. GA is closely associated with metabolic syndrome and decreased renal function, and the most important predictor of GA is hyperuricemia. There is a bidirectional relationship between hyperuricemia and renal function: While decreased renal function can cause hyperuricemia, hyperuricemia can also lead to renal failure by causing damage, scarring, and stone formation through endothelial dysfunction, activation of the renin-angiotensin system, hypertension, and accumulation of uric acid crystals in the kidneys [13,14]. Another investigation highlighted that GDF-15 may serve as an indicator associated with renal health and function across a broad population group [15].

Therefore, this study aimed to determine serum GDF-15 levels in gouty arthritis and to evaluate the relationship between kidney function and GDF-15.

## 2. Materials and Methods

### 2.1. Study Design and Participants

This research was designed as a forward-looking, observational study with a case–control structure. It was carried out in the rheumatology clinic from March 2024 to January 2025 by a specialist in the field. The aim was to explore the association between blood levels of GDF-15 and indicators of kidney function and systemic inflammation in individuals identified with gouty arthritis, based on the classification standards established by the American College of Rheumatology and the European League Against Rheumatism in 2015 [16]. The study included 60 patients with gouty arthritis and 60 healthy controls, matched for gender and body mass index (BMI). The gout group included individuals in the intercritical (n = 30) and acute attack (n = 30) phases of the disease. Participants were enrolled consecutively during routine clinical visits throughout the study period.

The study was conducted in accordance with the ethical standards outlined in the Declaration of Helsinki [17] and received approval from the relevant institutional ethics committee (Approval No: 2024/47). Before participation, all subjects provided written informed consent.

### 2.2. Eligibility Criteria

Study subjects were selected using specific inclusion and exclusion criteria to maintain a uniform study group and enhance the reliability of the results. Eligible participants were adults aged 18 years or older with a clinically and laboratory-confirmed diagnosis of gout. The diagnosis adhered to the classification standards established by the American College of Rheumatology and the European League Against Rheumatism in 2015 [16]. Patients were enrolled consecutively during intercritical periods or acute gout attacks as recorded by the rheumatology outpatient clinic. Healthy controls were BMI- and sex-matched individuals confirmed by a comprehensive history, physical examination, laboratory tests, and imaging. Exclusion criteria include individuals who are smokers or drinkers, have autoimmune disease, endocrine diseases, a cancer history, or other rheumatologic diseases; those receiving immunosuppressive therapy (like steroids); patients with acute cardiovascular diseases, hepatic failure, acute and chronic infection. All participants underwent a comprehensive clinical evaluation (blood pressure, arterial measurements, anamnesis, and physical examination). Demographic and anthropometric data, including age, sex distribution, height, weight, and BMI, were recorded. Additionally, comorbid conditions, medication use, and habits (smoking and alcohol use) were documented. A total of 15 participants reported a history of alcohol use, including six individuals in the control group, five in the intercritical gout group, and four in the acute gout attack group. However, none of the participants were current alcohol users at the time of blood sample collection.

#### Medication Usage Profile

Medication usage profiles revealed that 45% of patients used colchicine, 40% were prescribed allopurinol, and 3.3% used febuxostat. Regarding anti-inflammatory treatments, 10% of patients used NSAIDs. Among cardiovascular-related medications, 33.3% used ACE inhibitors, 26.7% used ARB inhibitors, 31.7% used calcium channel blockers, and 30% used beta blockers. For the management of dyslipidemia, 25% of patients were on statins, 3.3% were on fenofibrate, and 71.7% did not use any anti-hyperlipidemic agents. Some of the participants were using medications that may influence GDF-15 levels, such as ACE inhibitors, ARBs, statins, or beta-blockers. While this information was recorded and reported, statistical adjustments for these medications were not performed due to the limited sample size. This is acknowledged as a limitation in interpreting the biomarker levels.

### 2.3. Sample Collection and Laboratory Analyses

#### Blood Sample Collection

To minimize the effects of daily biological variations on laboratory results, venous blood samples were drawn from participants early in the morning following a minimum of eight hours of fasting. Metabolic and kidney function were assessed through measurements including fasting blood sugar, blood urea nitrogen (BUN), creatinine, estimated glomerular filtration rate (eGFR), and serum uric acid levels. The lipid panel consisted of total cholesterol (TC), triglycerides, as well as concentrations of high- and low-density lipoproteins (HDL and LDL). Markers of inflammation were evaluated by measuring levels of CRP and ESR. All biochemical analyses were performed using an autoanalyzer at the Clinical Biochemistry Laboratory of Recep Tayyip Erdoğan University Training and Research Hospital.

### 2.4. Measurement of Serum GDF15 Levels

The levels of GDF-15 in serum samples were measured using a commercially sourced enzyme-linked immunosorbent assay (ELISA) kit (Quantikine ELISA, R&D Systems, Minneapolis, MN, USA; Catalog No: DGD150). Samples previously stored at −80 °C were gently thawed at room temperature and diluted according to the manufacturer’s recommendations. All assays were performed in duplicate to ensure reproducibility. In brief, 100 µL of standards, controls, or serum samples were dispensed into 96-well plates pre-coated with monoclonal antibodies specific to human GDF15 and incubated on a microplate shaker at room temperature for 2 h. After washing the wells four times to remove unbound material, 200 µL of conjugated detection reagent was added to each well and incubated for 1 h. Following an additional wash step, 200 µL of substrate solution were added, and the reaction was allowed to proceed for 30 min in the dark. The enzymatic reaction was then stopped by the addition of 50 µL of stop solution. Absorbance values were measured at 450 nm using a microplate reader, with background correction performed at either 540 or 570 nm. GDF15 concentrations were interpolated from a standard curve generated using recombinant GDF15, based on a four-parameter logistic (4-PL) regression model.

### 2.5. Statistical Analysis

All statistical analyses were carried out using IBM SPSS Statistics for Windows, Version 23.0 (IBM Corp., Armonk, NY, USA). The distribution of continuous variables was examined using the Shapiro–Wilk test. Since the assumption of normality was not satisfied, non-parametric methods were employed. Comparisons involving more than two groups were conducted using the Kruskal–Wallis test. For variables with significant results in the Kruskal–Wallis test, post hoc pairwise comparisons were performed using the Bonferroni-adjusted Mann–Whitney U test. For variables that demonstrated significant differences between groups, correlation analyses were performed separately within the intercritical gout and acute gout attack subgroups. Within these subgroups, the associations between serum GDF-15 levels and selected clinical, biochemical, or hematological parameters were assessed using Spearman’s rank correlation coefficient. Additionally, two separate multiple linear regression analyses were performed to explore independent associations: one with eGFR as the dependent variable and GDF-15, age, sex, CRP, and ESR as predictors, and another with GDF-15 as the dependent variable and the same predictors included. Variables with *p* < 0.05 were considered statistically significant. To assess the diagnostic utility of GDF-15 in distinguishing between patients and healthy controls, receiver operating characteristic (ROC) curve analysis was employed. The area under the curve (AUC), 95% confidence intervals (CI), and optimal cut-off values derived from the Youden index were reported. Continuous variables were expressed as medians and interquartile ranges (IQR), while categorical variables were presented as frequencies and percentages. A *p*-value of less than 0.05 (two-tailed) was considered statistically significant. All tables were prepared using Microsoft Excel (Microsoft Corp., Redmond, WA, USA), and figures were generated using IBM SPSS and OriginPro 2024 (OriginLab Corp., Northampton, MA, USA).

## 3. Results

### 3.1. Comparison of Demographic and Clinical Data

The baseline demographic and clinical characteristics of the study groups are presented in Table 1. The proportion of female participants was similar across the groups (control: 25.0%, acute gout attack: 23.3%, and intercritical gout: 30.0%; *p* > 0.05). There were no significant differences among groups in terms of BMI (*p* = 0.086). The frequency of comorbid conditions was significantly higher in both gout groups compared to the control group (all *p* < 0.05). The median age was significantly higher in both the acute gout attack group (67.5 years) and the intercritical gout group (60 years) compared to the control group (53 years) (*p* = 0.001).

### 3.2. Renal Function and Uric Acid Levels

Significant differences were detected among the study groups in renal function and uric acid profiles (Table 2). The acute gout attack and intercritical gout groups exhibited elevated creatinine, BUN, and uric acid levels, as well as a decreased eGFR, compared to the control group (all *p* = 0.001). Furthermore, eGFR was significantly lower in the acute gout attack group than in the intercritical gout group.

### 3.3. Serum GDF-15 Levels

Figure 1 illustrates that patients experiencing an acute gout flare and those in the intercritical phase exhibited notably elevated serum GDF-15 levels compared to the control group (*p* < 0.001). Furthermore, the acute gout attack group showed significantly higher GDF-15 concentrations than the intercritical gout group (*p* < 0.01).

### 3.4. Biochemical Parameters

Biochemical parameters among the study groups are presented in Table 3. Serum albumin and HDL cholesterol levels were significantly lower in both the acute gout attack and intercritical gout groups compared to the control group. CRP levels were markedly elevated in the acute gout attack group, while intercritical gout patients had intermediate values. No significant differences were observed among the groups in terms of glucose, total cholesterol, triglyceride, or LDL cholesterol levels (*p* > 0.05).

### 3.5. Hematological and Inflammatory Parameters

Marked differences were observed in hematological parameters among the groups (Table 4). Both gout groups exhibited higher WBC, ESR, NE, and NLR values compared to the controls, with the most pronounced elevations in the acute gout attack group (WBC, *p* = 0.014; ESR, *p* = 0.001; NEU, *p* = 0.004; NLR, *p* = 0.005). MON and IMG were also significantly increased in the acute gout group (MON *p* = 0.034 and IMG *p* = 0.04). In contrast, no significant group differences existed for LYM, Hb, PLT, MLR, PLR, or SII.

### 3.6. GDF-15 and Lab Correlations in Gout Phases

The correlations between GDF-15 and clinical, biochemical, and hematological parameters were assessed separately in the acute gout attack and intercritical gout groups (Figure 2 and Figure 3). GDF-15 showed moderate correlations with renal function markers in both groups. In the acute gout attack group, moderate positive correlations were observed with urea (r = 0.52, *p* < 0.01) and creatinine (r = 0.53, *p* < 0.01), while a moderate negative correlation was found with eGFR (r = −0.44, *p* < 0.05). In the intercritical gout group, GDF-15 was moderately and negatively correlated with eGFR (r = −0.53, *p* < 0.01), weakly and negatively correlated with albumin (r = −0.37, *p* < 0.05), and weakly and positively correlated with HDL cholesterol (r = 0.40, *p* < 0.05). For both groups, correlations between GDF-15 and other biochemical or hematological parameters were weak and not significant.

### 3.7. Multiple Regression Analysis

In the first regression model, eGFR was set as the dependent variable, and GDF15, age, sex, CRP, and ESR were included as predictors. Among these, only GDF15 emerged as a significant independent predictor of reduced eGFR (β = −0.0187, *p* < 0.001), while age (*p* = 0.55), sex (*p* = 0.15), CRP (*p* = 0.54), and ESR (*p* = 0.26) did not show significant associations. In the second model, where GDF15 was used as the dependent variable, several predictors were found to be substantial. Specifically, lower eGFR (β = −17.08, *p* < 0.001), higher CRP (β = 20.86, *p* = 0.027), higher ESR (β = 10.57, *p* = 0.014), and male sex (β = 234.22, *p* = 0.016) were independently associated with elevated GDF15 levels. Age did not reach statistical significance in this model (*p* = 0.061).

### 3.8. Evaluation of the Diagnostic Performance of GDF-15, CRP, and ESR

ROC curve analyses revealed that serum GDF-15 had excellent diagnostic accuracy in discriminating between patients with acute gout attacks and healthy controls, yielding an area under the curve (AUC) of 0.980 (95% CI: 0.945–1.000, *p* < 0.0001). CRP (AUC = 0.877, 95% CI: 0.798–0.956, *p* < 0.0001) and ESR (AUC = 0.821, 95% CI: 0.728–0.915, *p* < 0.0001) demonstrated good discriminatory capacities but were inferior compared to GDF-15. The optimal cut-off value identified for GDF-15 was 1160.8 pg/mL, corresponding to a sensitivity of 96.7% and a specificity of 93.3% (Figure 4).

For intercritical gout, ROC analysis similarly indicated excellent discriminative ability for serum GDF-15 (AUC = 0.966, 95% CI: 0.912–1.020, *p* < 0.0001). In contrast, CRP did not exhibit statistically significant diagnostic performance (AUC = 0.578, 95% CI: 0.430–0.726, *p* = 0.229), while ESR demonstrated modest yet statistically significant discriminatory ability (AUC = 0.651, 95% CI: 0.529–0.772, *p* = 0.020). For distinguishing intercritical gout patients, the optimal GDF-15 threshold was 1147.5 pg/mL, achieving sensitivity and specificity values of 93.3% (Figure 5). Collectively, these findings underscore the robust and consistent diagnostic potential of GDF-15 across different phases of gout.

## 4. Discussion

To our knowledge, this research is the first to report an association between serum GDF-15 concentrations and gouty arthritis. The findings of our study show that individuals diagnosed with GA have markedly increased GDF-15 levels in their blood relative to healthy subjects, with peak concentrations found during both the active and symptom-free periods of the condition. These results suggest a potential connection between GDF-15 and the inflammatory mechanisms underlying gouty arthritis. ROC analysis revealed that GDF-15 exhibits high diagnostic accuracy in distinguishing gout patients from healthy controls; however, its limited ability to differentiate between acute attacks suggests that it should be used in conjunction with other clinical and laboratory markers. Given this limitation, GDF-15 may not be sufficiently specific on its own and should be interpreted in conjunction with traditional inflammatory markers to enhance diagnostic precision. Classic inflammatory markers, such as CRP, WBC, NEU, and ESR, were significantly elevated in patients experiencing acute gout attacks.

These findings align with recent research on persistent inflammatory conditions. For example, higher concentrations of GDF-15 in the blood have been observed in individuals with rheumatoid arthritis, where its levels correlate positively with the disease’s intensity and progression, highlighting its potential as a diagnostic marker [18]. In a related study, Xu WD and colleagues reported that individuals diagnosed with systemic lupus erythematosus exhibited elevated serum GDF-15 concentrations compared to healthy subjects. These elevated levels were positively linked to disease activity and inversely associated with complement protein levels. Moreover, administering GDF-15 to lupus-prone mice led to a decrease in several proinflammatory cytokines, including IL-1β, IL-18, and IL-22, highlighting its role in modulating the immune response [19]. Taken together, the evidence indicates that GDF-15 could play a similar role in the development of GA. Moreover, decreased HDL levels in gout groups may indicate chronic inflammation and metabolic dysfunction.

GA and rheumatoid arthritis (RA) are both inflammatory joint diseases, but their underlying mechanisms are different [20]. GA arises from the activation of the innate immune system, primarily through the NLRP3 inflammasome complex, which is triggered by monosodium urate crystals [21]. This leads to a rapid increase in inflammatory cytokines such as IL-1β [22]. In contrast, RA is a chronic autoimmune disease characterized by adaptive immune responses and the production of cytokines such as TNF-α and IL-6 [23]. Although elevated GDF-15 levels have been reported in RA and other inflammatory diseases [24,25], our study is the first to demonstrate a significant rise in serum GDF-15 levels in gouty arthritis, particularly during acute attacks, suggesting that this cytokine may reflect a distinct acute-phase response in GA.

Our findings reveal a significant link between GDF-15 levels and indicators of kidney function in patients with gouty arthritis. This aligns with earlier research reported in the literature. For instance, Macedo and colleagues identified a positive relationship between GDF-15 and serum creatinine in individuals with lupus nephritis [26]. Higher GDF-15 concentrations have been linked to chronic kidney disease, microalbuminuria, and accelerated loss of kidney function in individuals suffering from diabetic nephropathy [27]. Research on IgA nephropathy and idiopathic membranous nephropathy has shown that increased serum levels of GDF-15 are closely linked to impaired kidney function and disease progression [28,29]. In our cohort, serum GDF-15 levels showed a positive association with creatinine and an inverse relationship with eGFR, indicating its potential role as an indicator of kidney dysfunction in patients with GA.

GDF15 is a stress-induced cytokine with notable anti-inflammatory properties, partly through the inhibition of NF-κB signaling by suppressing TGF-β-activated kinase 1 [30]. In lipopolysaccharide-induced inflammatory models, growth differentiation factor 15 significantly reduces the expression and secretion of proinflammatory cytokines, including IL-1β, IL-6, and tumor necrosis factor-alpha [31]. Given that monosodium urate crystals drive gouty inflammation primarily through NLRP3 inflammasome activation and the subsequent release of IL-1β and IL-18 [32], the suppressive effects of GDF15 on these mediators suggest a potential regulatory role in modulating gout-associated inflammatory responses.

Although the cross-sectional design of our study does not permit direct causal inference, the strong and independent association observed between elevated serum GDF-15 levels and decreased eGFR may reflect more than a coincidental correlation. Emerging evidence suggests that GDF-15 is not merely a passive biomarker but may actively participate in the regulation of renal injury and stress responses [33]. It has been demonstrated to exert protective effects in renal tissue by modulating and attenuating oxidative stress, as well as inhibiting proinflammatory signaling pathways [34]. Therefore, elevated GDF-15 levels in our patients may indicate both a response to renal dysfunction and a compensatory mechanism aimed at limiting further tissue damage. Further longitudinal and experimental studies are warranted to determine whether GDF-15 plays a causal role in preserving renal function in the context of gout-related systemic inflammation.

Under normal physiological conditions, GDF-15 circulates at low levels in the serum, but its levels increase rapidly in response to inflammatory states, ischemia, oxidative stress, or hypoxia [35]. In acute kidney injury, GDF-15 levels rise quickly and have been shown to protect renal tubular cells against cytotoxic damage. In chronic kidney injury, GDF-15 suppresses proinflammatory macrophages and activates anti-inflammatory macrophages, thereby contributing to renal protection. It has also been demonstrated to increase Klotho expression in renal tubules, thereby delaying cellular senescence, fibrosis, and loss of kidney function [36,37]. Additionally, GDF-15 decreases the production of adhesion proteins on the surface of blood vessel lining cells, thereby restricting the movement of white blood cells and preserving the integrity of the glomerular filtration barrier, which helps prevent proteinuria [37,38]. In gouty arthritis, hyperuricemia and persistent inflammation can lead to tubulointerstitial nephritis, oxidative stress, endothelial dysfunction, urate nephropathy, decreased glomerular filtration rate, and progression to chronic kidney disease (CKD) through fibrotic mechanisms [39,40]. In this context, the connection between elevated GDF-15 concentrations, diminished renal function, and inflammatory activity in patients with GA supports a potential protective or compensatory role for GDF-15 in the progression of kidney disease. In addition, despite its anti-inflammatory effects, persistently high levels may indicate loss of glomerular and tubular function. Therefore, GDF-15 is a potential biomarker for assessing renal involvement in patients with gout.

## 5. Conclusions

In this study, serum GDF-15 levels were elevated in patients with GA and were associated with impaired renal function. GDF-15 may be a helpful biomarker reflecting disease activity and renal involvement in gouty arthritis. However, further studies in more extensive and diverse populations are needed to confirm these findings and clarify the clinical utility of GDF-15 in this context.

### Limitations

This research has some limitations. Primarily, the number of participants was limited, and since the study took place at only one location, the results may not apply to broader populations. Secondly, although the study groups showed notable age differences, correlation analysis demonstrated only a minimal relationship between age and serum GDF-15 levels. Third, the cross-sectional design does not allow for assessment of temporal changes or causal relationships. Finally, potential confounding factors such as concomitant medications, comorbidities, and dietary factors may have influenced the results. Specifically, medications known to affect GDF-15 levels, such as ACE inhibitors, statins, and beta-blockers, could not be statistically adjusted for due to the limited sample size.

## Figures and Tables

**Figure 1 biomedicines-13-01767-f001:**
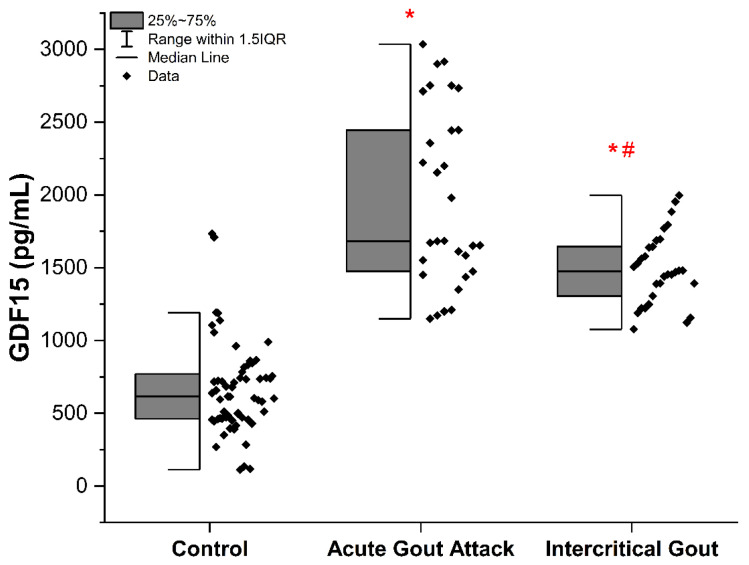
Serum GDF-15 levels across study groups. Pairwise analyses were performed using the Mann–Whitney U test. The differences marked with an asterisk (*) were significant compared to the control group, and those marked with a hash (#) were significant compared to the acute gout attack group (*p* < 0.05). GDF: growth differentiation factor 15.

**Figure 2 biomedicines-13-01767-f002:**
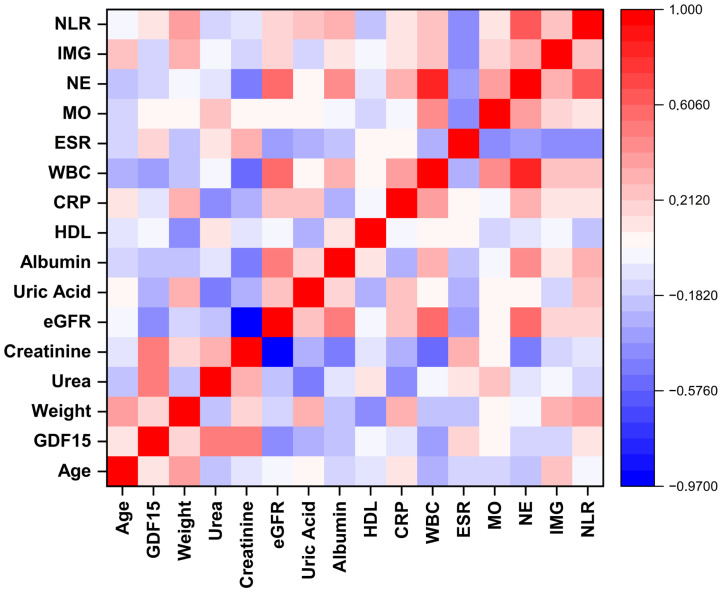
Spearman correlation matrix of demographic, biochemical, and hematological parameters in the acute gout attack group. NLR: neutrophil to lymphocyte ratio; IMG: immature granulocyte; NEU: neutrophil; MON: monocyte; ESR: erythrocyte sedimentation rate; WBC: white blood cell count; CRP: C-reactive protein; HDL: high-density lipoprotein; eGFR: estimated glomerular filtration rate; GDF-15: growth differentiation factor 15.

**Figure 3 biomedicines-13-01767-f003:**
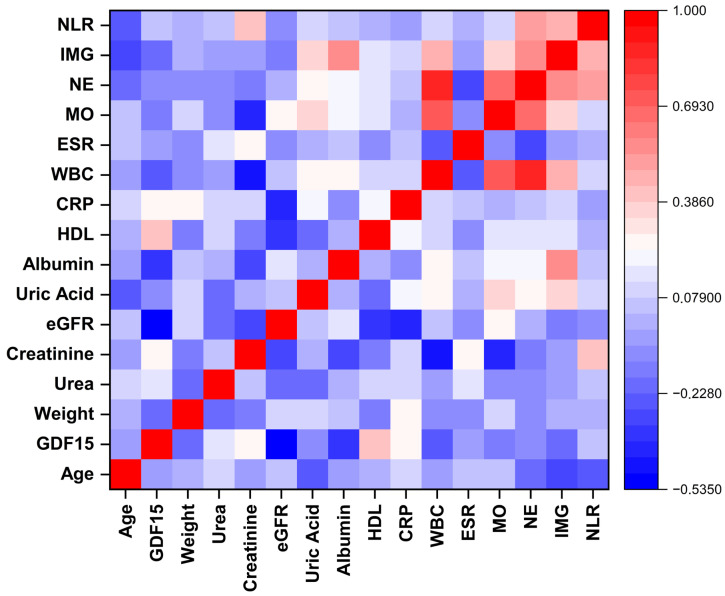
Spearman correlation matrix of demographic, biochemical, and hematological parameters in the intercritical gout group. NLR: neutrophil to lymphocyte ratio; IMG: immature granulocyte; NEU: neutrophil; MON: monocyte; ESR: erythrocyte sedimentation rate; WBC: white blood cell count; CRP: C-reactive protein; HDL: high-density lipoprotein; eGFR: estimated glomerular filtration rate; GDF-15: growth differentiation factor 15.

**Figure 4 biomedicines-13-01767-f004:**
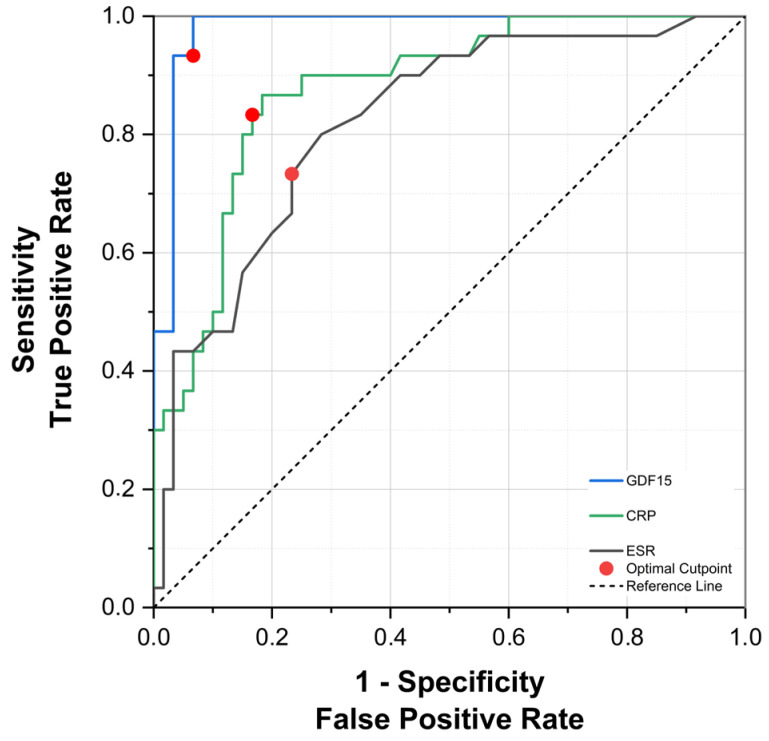
ROC curve for serum GDF-15, CRP, and ESR in the diagnosis of acute gout attack.

**Figure 5 biomedicines-13-01767-f005:**
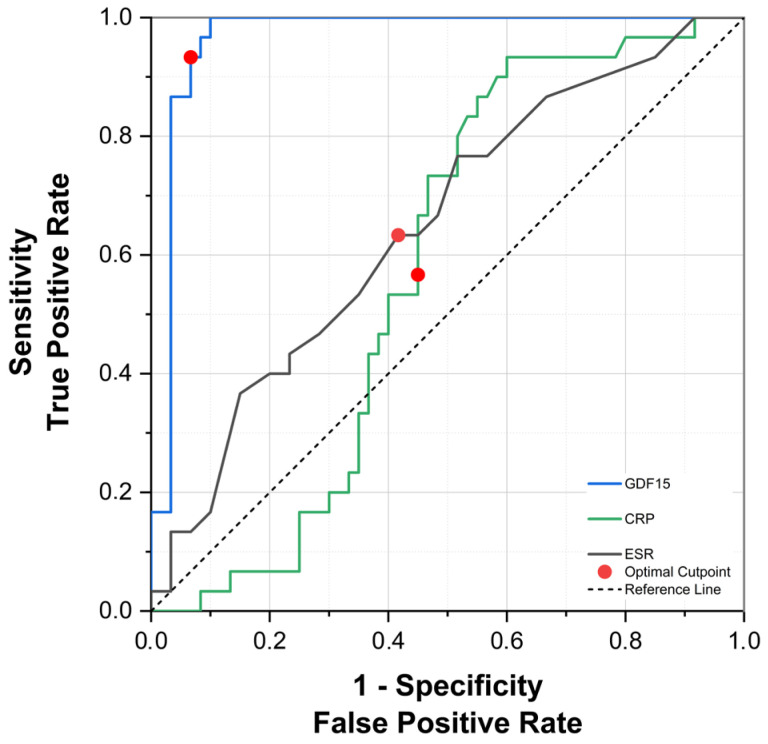
ROC curve for serum GDF-15, CRP, and ESR in the diagnosis of intercritical gout.

**Table 1 biomedicines-13-01767-t001:** Baseline demographic and clinical characteristics of study groups.

Parameters	Control (n = 60) Median (IQR)	Acute Gout Attack (n = 30) Median (IQR)	Intercritical Gout (n = 30) Median (IQR)	*p*-Value (Kruskal–Wallis/Chi-Square Test)
Sex (% Female, % Male, n)	25% (15), 75% (45)	23.3% (7), 76.7% (23)	30% (9), 70% (21)	*p* > 0.05 ^1^
Age (Years)	53 (10.5)	67.5 * (18.5)	60 * (18.5)	*p* = 0.001 ^2^
Height (cm)	167.5 (9)	170.5 (10)	170 (9)	*p* = 0.275 ^2^
Weight (kg)	80.5 (11.7)	86.5 * (10)	85 * (9.5)	*p* = 0.025 ^2^
BMI (kg/m^2^)	27.7 (5.2)	28.9 (6.84)	30.4 (4.52)	*p* = 0.086 ^2^
Hyperlipidemia (%/n)	10% (6)	46.6% (14)	38.4% (10)	*p* < 0.05 ^1^
Coronary artery disease (%/n)	8.3% (5)	29.4% (10)	30.7% (8)	*p* < 0.05 ^1^
Heart failure (%/n)	-	17.6 (6)	19.2% (5)	
Chronic kidney disease (%/n)	-	11.7% (5)	15.3% (4)	
Diabetes mellitus (%n)	-	8.82% (3)	19.2% (5)	
Hypertension (%/n)	21.6% (13)	8.82% (3)	15.3% (4)	*p* = 0.32 ^1^

^1^ *p*-values calculated using the Chi-square test. ^2^ *p*-values calculated using the Kruskal–Wallis test. Where the Kruskal–Wallis test indicated significance (*p* < 0.05), pairwise comparisons were performed using the Mann–Whitney U test. * Statistically significant difference compared to the control group (*p* < 0.05).

**Table 2 biomedicines-13-01767-t002:** Renal function and uric acid profiles across study groups.

Parameters	Control (n = 60) Median (IQR)	Acute Gout Attack (n = 30) Median (IQR)	Intercritical Gout (n = 30) Median (IQR)	*p*-Value (Kruskal–Wallis)
Creatinine (mg/dL)	0.77 (0.2)	1.04 * (0.37)	0.92 * (0.29)	*p* = 0.001
BUN (mg/dL)	29 (14)	43.5 * (24.75)	35 * (18.5)	*p* = 0.001
eGFR (mL/min/1.73 m^2^)	97 (20.7)	68.5 * (34.5)	80 *^,#^ (24.5)	*p* = 0.001
Uric Acid (mg/dL)	4.95 (1.8)	7.25 * (2.95)	6.7 * (2.2)	*p* = 0.001

*p*-values calculated using the Kruskal–Wallis test. Where the Kruskal–Wallis test indicated significance (*p* < 0.05), pairwise comparisons were performed using the Mann–Whitney U test. The differences marked with an asterisk (*) were significant compared to the control group, and those marked with a hash (#) were significant compared to the acute gout attack group (*p* < 0.05). BUN: blood urea nitrogen; eGFR: estimated glomerular rate.

**Table 3 biomedicines-13-01767-t003:** Comparison of biochemical parameters among study groups.

Parameters	Control (n = 60) Median (IQR)	Acute Gout Attack (n = 30) Median (IQR)	Intercritical Gout (n = 30) Median (IQR)	*p*-Value (Kruskal–Wallis)
Glucose (mg/dL)	94.5 (12.5)	98.5 (19.7)	94 (14)	*p* = 0.280
Albumin (g/L)	46.1 (3.2)	43.5 * (4.4)	45 * (4.8)	*p* = 0.002
Cholesterol (mg/dL)	215 (53.2)	222.5 (59.2)	203 (75.5)	*p* = 0.570
Triglyceride (mg/dL)	124 (92.4)	148.5 (94)	151 (92)	*p* = 0.073
HDL (mg/dL)	53.7 (16.9)	47.8 * (10.9)	44.5 * (14.9)	*p* = 0.004
LDL (mg/dL)	141 (47)	145.5 (47.5)	134 (43.5)	*p* = 0.492
CRP (mg/L)	1.7 (3.9)	8.8 * (9.2)	2.51 ^#^ (1.51)	*p* = 0.001

Group comparisons were initially performed using the Kruskal–Wallis test. For parameters with a significant group effect (*p* < 0.05), pairwise comparisons were conducted using the Mann–Whitney U test. Differences marked with an asterisk (*) are significant versus the control group, and those marked with a hash (#) are significant versus the acute gout attack group (*p* < 0.05). HDL: high-density lipoprotein; LDL: low-density lipoprotein; CRP: C-reactive protein.

**Table 4 biomedicines-13-01767-t004:** Hematological parameter comparison between groups.

Parameters	Control (n = 60) Median (IQR)	Acute Gout Attack (n = 30) Median (IQR)	Intercritical Gout (n = 30) Median (IQR)	*p*-Value (Kruskal–Wallis)
WBC (10^3^/µL)	6.5 (2)	7.35 * (3.34)	6.63 ^#^ (2.2)	*p* = 0.014
ESR (mm/1 h)	8 (9)	17 * (16.5)	12 ^#^ (12)	*p* = 0.001
LYM (10^3^/µL)	2.1 (0.7)	2.23 (1.05)	2.34 (0.96)	*p* = 0.482
MON (10^3^/µL)	0.4 (0.1)	0.49 * (0.16)	0.44 (0.18)	*p* = 0.034
NEU (10^3^/µL)	3.7 (1.8)	4.34 * (2.28)	3.59 ^#^ (1.32)	*p* = 0.004
Hb (g/dL)	14 (1.9)	14.2 (2.2)	14.9 (2.2)	*p* = 0.114
PLT (10^3^/µL)	259.5 (90)	243 (67)	241 (61)	*p* = 0.657
IMG (10^3^/µL)	0.01 (0.01)	0.02 * (0.02)	0.01 ^#^ (0.01)	*p* = 0.04
NLR	1.7 (1.02)	2.2 * (1.09)	1.51 ^#^ (0.78)	*p* = 0.005
MLR	0.19 (0.11)	0.23 (0.14)	0.21 (0.1)	*p* = 0.081
PLR	119.5 (55.03)	113.9 (61.03)	103.3 (47.5)	*p* = 0.377
SII	46.1 (132.3)	59.1 (34.6)	45.6 (29.1)	*p* = 0.380

Group comparisons were initially performed using the Kruskal–Wallis test. For parameters with a significant group effect (*p* < 0.05), pairwise comparisons were conducted using the Mann–Whitney U test. Differences marked with an asterisk (*) are significant versus the control group, and those marked with a hash (#) are significant versus the acute gout attack group (*p* < 0.05). HDL: high-density lipoprotein; LDL: low-density lipoprotein; CRP: C-reactive protein; WBC: white blood cell; ESR: erythrocyte sedimentation rate; LYM: lymphocyte; MON: monocyte; NEU: neutrophil; Hb: hemoglobin; PLT: platelet; IMG: immature granulocyte; NLR: neutrophil to lymphocyte ratio; MLR: monocyte to lymphocyte ratio; PLR: platelet to lymphocyte ratio; SII: systemic inflammatory index.

## Data Availability

The original contributions presented in this study are included in the article; further inquiries can be directed to the corresponding author.

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
