# Peer review of "GDF-15 Levels in Gouty Arthritis and Correlations with Decreasing Renal Function: A Clinical Study"

_biomedicines, 2025, doi:10.3390/biomedicines13071767_

Round 1
Reviewer 1 Report
Comments and Suggestions for Authors
Dear AUTHORS, interesting report about GDF-15 in GA. Briefly, do you have information about the medications that the patients have been using in the acute event or in the intercritical moments? Are these drug medications associated with steroids, NSAIDs, urate lowering therapies (please specify)? Some information about the use or abuse of alcohol? Clinically, in the acute events which was the main manifestation, are these patients free of thophus? Do you have more information about the relationship between GDF-15 with proinflammatory cytokines in the context of gout (like IL-1, IL-6), this could be added in discussion section. Best regards.
Author Response
Author’s Response for Reviewer-1
We sincerely thank Reviewer 1 for their thoughtful and constructive feedback on our manuscript. The comments provided were instrumental in improving the scientific clarity, methodological transparency, and overall quality of the study. In response, we have made several important revisions. These include the addition of detailed medication usage profiles, clarification on alcohol history, expanded methodological limitations, and a newly integrated mechanistic discussion of GDF15’s role in inflammatory cytokine modulation based on relevant literature. We believe that these revisions have significantly enhanced the manuscript and comprehensively addressed the reviewer’s concerns.
Reviewer-1
Comment 1: Do you have information about the medications that the patients have been using in the acute event or in the intercritical moments? Are these drug medications associated with steroids, NSAIDs, urate lowering therapies (please specify)?
Response 1: Thank you for highlighting this important issue regarding medication use. In response, we have revised the Materials and Methods section to include a detailed summary of the medication profiles of our study population. Specifically, 45% of patients were using colchicine, 40% were prescribed allopurinol, and 3.3% used febuxostat. Additionally, 10% used NSAIDs. Information on cardiovascular and lipid-lowering medications was also included, such as ACE inhibitors, ARBs, calcium channel blockers, beta-blockers, statins, and fenofibrate. These data were collected during structured clinical interviews and were taken into account in the interpretation of serum GDF-15 levels. However, due to the limited sample size, we did not perform separate statistical adjustments for the effects of individual medications. We acknowledge this as a limitation and have noted it in the Discussion section.
Comment 2: Some information about the use or abuse of alcohol?
Response 2: We appreciate the reviewer’s comment regarding alcohol use. In the revised manuscript, we have clarified that none of the participants were active alcohol users at the time of enrollment. However, 15 individuals across all study groups reported a history of past alcohol use (6 in the control group, 5 in the intercritical gout group, and 4 in the acute gout attack group). This clarification has been added to the “Eligibility Criteria” section.
Comment 3: Clinically, in the acute context, are these patients under other medications that might influence the markers?
Response 3: We appreciate the reviewer's thoughtful observation. In the revised manuscript, we have provided a detailed description of the medication profiles of the study participants. Several patients were indeed using medications that have been previously reported to influence serum GDF-15 levels, including ACE inhibitors, ARBs, statins, and beta-blockers. While the use of these agents was recorded and transparently reported in the Methods section, we were unable to statistically adjust for their potential confounding effects due to the limited sample size. This methodological limitation has now been explicitly acknowledged in the revised version, both in the Materials and Methods and in the Limitations section of the manuscript.
Comment 4: It is possible to add more information about the relationship between GDF-15 with proinflammatory cytokines in the context of gout (like IL-1, IL-6), this could be added in discussion section.
Response 4: Thank you for this insightful comment. In response, we have revised the discussion section to provide a more comprehensive overview of the potential anti-inflammatory mechanisms of growth differentiation factor 15 (GDF15), supported by recent experimental findings.
Specifically, we added the following paragraph to contextualize our findings within current mechanistic evidence:
“GDF15 is a stress-induced cytokine with notable anti-inflammatory properties, partly through the inhibition of NF-κB signaling by suppressing transforming growth factor-beta-activated kinase 1 (Ratnam et al., 2020). In lipopolysaccharide-induced inflammatory models, growth differentiation factor 15 significantly reduces the expression and secretion of proinflammatory cytokines, including IL-1β, IL-6, and tumor necrosis factor-alpha (Song et al., 2021). Given that monosodium urate crystals drive gouty inflammation primarily through NLRP3 inflammasome activation and the subsequent release of IL-1β and IL-18 (Kim et al., 2022), the suppressive effects of GDF15 on these mediators suggest a potential regulatory role in modulating gout-associated inflammatory responses.”
This addition strengthens the biological plausibility of our findings by linking the observed increase in GDF15 levels during acute gout attacks to known anti-inflammatory signaling pathways and cytokine suppression mechanisms described in the literature. We hope this provides greater clarity and depth to the discussion.
Ratnam, N. M., Peterson, J. M., Talbert, E. E., Ladner, K. J., Rajasekera, P. V., Schmidt, C. R., Dillhoff, M. E., Swanson, B. J., Haverick, E., Kladney, R. D., Williams, T. M., Leone, G. W., Wang, D. J., & Guttridge, D. C. (2017). NF-κB regulates GDF-15 to suppress macrophage surveillance during early tumor development. The Journal of clinical investigation, 127(10), 3796–3809. https://doi.org/10.1172/JCI91561
Song, H., Chen, Q., Xie, S., Huang, J., & Kang, G. (2020). GDF-15 prevents lipopolysaccharide-mediated acute lung injury via upregulating SIRT1. Biochemical and biophysical research communications, 526(2), 439–446. https://doi.org/10.1016/j.bbrc.2020.03.103
Kim S. K. (2022). The Mechanism of the NLRP3 Inflammasome Activation and Pathogenic Implication in the Pathogenesis of Gout. Journal of rheumatic diseases, 29(3), 140–153. https://doi.org/10.4078/jrd.2022.29.3.140
Osman CURE (M.D.)
On behalf of the Authors
Reviewer 2 Report
Comments and Suggestions for Authors
This manuscript explores the changes of serum GDF-15 levels in patients with gouty arthritis (GA) and its correlation with renal function by a prospective case-control research design, which has good innovation and clinical significance. The results showed that GDF-15 was significantly increased in patients with GA, especially during the acute attack period, and was associated with the decline of renal function. However, the following key issues need to be addressed for enhancing the integrity of the research and its clinical application value.
Major:
- The total sample size was relatively small (n=60 patients /60 controls), and the age of the GAgroup was significantly higher than that of the control group (67.5/60 years vs. 53 years), which might introduce age-related confounding bias. Suggesting to provide multivariate correction results.
- GDF-15 is elevated in various diseases such as cardiovascular diseases, tumors, CKD, but thisstudy did not compare the GDF-15 levels of GA with those of other inflammatory joint diseases such as rheumatoid arthritis, and not analyse its incremental value relative to existing markers such as CRP, ESR. Suggesting to supplement the ROC curve analysis to compare the efficacy of GDF-15 and CRP/ESR (alone or in combination) in the diagnosis of GA.
- Suggesting to supplement the multivariate regression results of GDF-15 and renal function indicators, after adjusting for age, gender, and comorbidities.
- The association between GDF-15 and renal function was only based on correlation analysis, lacking the discussion of causal relationship.
Minor:
- The data of "hypertension" in Table 1 is contradictory, 21.6% in the control group vs. 8.82% in the acute group, but p<0.05. It is necessary to recheck the statistical method or data entry errors.
- It is necessary to have an in-depth discussion on the differences in inflammatory pathways between GA and RA by comparing the reported GDF-15 levels in RA patients.
- Please explain the clinical significance that GDF-15 was significantly higher in the acute phase than in the intermission phase, but the ROC cut-off value was close (1160.8 vs. 1147.5 pg/mL).
Author Response
Author’s Response for Reviewer-2
We sincerely thank Reviewer 2 for their thoughtful and constructive comments, which have significantly contributed to improving the scientific rigor, clarity, and clinical relevance of our manuscript. We carefully addressed all major and minor concerns raised, including adjustments for potential age-related confounding, comprehensive ROC analyses comparing GDF-15 with conventional biomarkers, expanded regression models adjusting for relevant covariates, and a deeper discussion of mechanistic insights and differential inflammatory pathways in gouty versus rheumatoid arthritis. Each comment has been addressed point by point, and the manuscript has been revised accordingly to enhance its methodological robustness and interpretative depth. Additionally, the English language of the article was professionally edited.
Reviewer-2
Comment 1: The total sample size was relatively small (n=60 patients /60 controls), and the age of the GA group was significantly higher than that of the control group (67.5/60 years vs. 53 years), which might introduce age-related confounding bias, suggesting the need to provide multivariate correction results.
Response 1: We thank the reviewer for highlighting the potential for age-related confounding, given the higher median age in the gout group compared to the controls. To address this concern, we performed a multiple linear regression analysis using eGFR as the dependent variable and GDF-15, age, sex, CRP, and ESR as predictors. The results demonstrated that GDF-15 was the only independent predictor of reduced eGFR (β = –0.0187, p < 0.001), while age did not show a statistically significant association (p = 0.547). This finding suggests that the observed association between GDF-15 and renal function is not confounded by age. We have included this analysis and its interpretation in the revised Results and Statistical Analysis sections of the manuscript.
Comment 2: GDF-15 is elevated in various diseases such as cardiovascular diseases, tumors, and CKD, but this study did not compare the GDF-15 levels of GA with those of other inflammatory joint diseases such as rheumatoid arthritis, and did not analyze its incremental value relative to existing markers such as CRP and ESR. Suggesting to supplement the ROC curve analysis to compare the efficacy of GDF-15 and CRP/ESR (alone or in combination) in the diagnosis of GA.
Response 2: We thank the reviewer for suggesting additional ROC analyses comparing the diagnostic performance of GDF-15 with commonly utilized inflammatory markers, CRP and ESR. We have conducted comprehensive ROC curve analyses separately for patients with acute gout attacks and those with intercritical gout (Results Section 3.8). The results demonstrate that serum GDF-15 exhibits outstanding diagnostic accuracy for acute gout attacks (AUC = 0.980), surpassing that of CRP (AUC = 0.877) and ESR (AUC = 0.821). Similarly, in intercritical gout, GDF-15 demonstrated excellent discriminatory performance (AUC = 0.966), whereas CRP showed limited diagnostic utility (AUC = 0.578), and ESR exhibited modest performance (AUC = 0.651). Figures 4 and 5 have been updated to reflect these additional analyses.
Comment 3: Suggesting to supplement the multivariate regression results of GDF-15 and renal function indicators, after adjusting for age, gender, and comorbidities.
Response 3: We appreciate the reviewer's important recommendation. In the revised manuscript, we included multivariate linear regression analyses to evaluate the relationship between serum GDF-15 levels and renal function indicators (eGFR), while adjusting for age, sex, CRP, and ESR. The results indicated that GDF-15 was an independent and significant negative predictor of eGFR (β = –0.0187, p < 0.001), whereas age, sex, CRP, and ESR were not significantly associated with eGFR. Furthermore, we constructed a reverse model using GDF-15 as the dependent variable, which revealed that lower eGFR, higher CRP and ESR levels, and male sex were independently associated with elevated GDF-15 concentrations. These results have been presented in the revised manuscript under the Results subsection 3.7, titled Multiple Regression Analysis.
Comment 4: The association between GDF-15 and renal function was only based on correlation analysis, lacking the discussion of causal relationship.
Response 4: We appreciate the reviewer’s insightful observation regarding the limitation of correlation analysis in establishing a causal relationship between serum GDF-15 levels and renal function decline. Although our study design is inherently cross-sectional and does not permit causal inference, we have now expanded the discussion to include mechanistic perspectives derived from recent experimental literature. Specifically, we highlight that GDF-15 is not merely a passive marker, but may be actively involved in regulating renal stress and controlling inflammation.
In the revised manuscript, the following paragraph was added to the Discussion section:
“Although the cross-sectional design of our study does not permit direct causal inference, the strong and independent association observed between elevated serum GDF-15 levels and decreased eGFR may reflect more than a coincidental correlation. Emerging evidence suggests that GDF-15 is not merely a passive biomarker but may actively participate in the regulation of renal injury and stress responses (Lasaad et al., 2024). It has been demonstrated to exert protective effects in renal tissue by modulating and attenuating oxidative stress, as well as inhibiting pro-inflammatory signaling pathways (Zhang et al., 2024). Therefore, elevated GDF-15 levels in our patients may indicate both a response to renal dysfunction and a compensatory mechanism aimed at limiting further tissue damage. Further longitudinal and experimental studies are warranted to determine whether GDF-15 plays a causal role in preserving renal function in the context of gout-related systemic inflammation.”
We hope this addition sufficiently addresses the reviewer’s concern and improves the mechanistic depth and clinical relevance of our interpretation.
Lasaad, S., & Crambert, G. (2024). GDF15, an Emerging Player in Renal Physiology and Pathophysiology. International journal of molecular sciences, 25(11), 5956. https://doi.org/10.3390/ijms25115956
Zhang, X., Wang, S., Chong, N., Chen, D., Shu, J., Sun, J., Sun, Z., Wang, R., Wang, Q., & Xu, Y. (2024). GDF-15 alleviates diabetic nephropathy via inhibiting NEDD4L-mediated IKK/NF-κB signalling pathways. International immunopharmacology, 128, 111427. https://doi.org/10.1016/j.intimp.2023.111427
Comment 5: The data of "hypertension" in Table 1 is contradictory, 21.6% in the control group vs. 8.82% in the acute group, but p < 0.05. It is necessary to recheck the statistical method or data entry errors.
Response 5: We appreciate the reviewer’s careful attention to the hypertension data. Upon rechecking, we identified an inconsistency in the previously reported p-value. After recalculating using the chi-square test based on the actual group distributions, the corrected p-value for the comparison of hypertension prevalence among the groups is 0.32. This updated result has been corrected in Table 1 accordingly. We thank the reviewer for helping us improve the accuracy of our data presentation.
Comment 6: It is necessary to have an in-depth discussion on the differences in inflammatory pathways between GA and RA by comparing the reported GDF-15 levels in RA patients.
Response 6: We appreciate the reviewer's valuable suggestion. In the revised Discussion section (subsection 3.8), we have added a comparative paragraph outlining the key differences between the inflammatory mechanisms of gouty arthritis (GA) and rheumatoid arthritis (RA). While GA is primarily driven by innate immune activation via the NLRP3 inflammasome and the rapid release of cytokines, such as IL-1β, RA is characterized by the chronic activation of the adaptive immune system and cytokines like TNF-α and IL-6. Although elevated serum GDF-15 levels have been previously reported in RA and other inflammatory diseases, our findings suggest that GDF-15 may also rise during acute gout attacks, potentially reflecting phase-specific inflammatory dynamics.
Firestein, G. S. (2025). Firestein and Kelley’s Textbook of Rheumatology (Twelfth edition.). Elsevier Inc.
Kingsbury, S. R., Conaghan, P. G., & McDermott, M. F. (2011). The role of the NLRP3 inflammasome in gout. Journal of inflammation research, 4, 39–49. https://doi.org/10.2147/JIR.S11330
Dinarello C. A. (2011). Interleukin-1 in the pathogenesis and treatment of inflammatory diseases. Blood, 117(14), 3720–3732. https://doi.org/10.1182/blood-2010-07-273417
Smolen, J. S., Aletaha, D., & McInnes, I. B. (2016). Rheumatoid arthritis. Lancet (London, England), 388(10055), 2023–2038. https://doi.org/10.1016/S0140-6736(16)30173-8
He, Y. W., & He, C. S. (2022). Association of Growth and Differentiation Factor 15 in Rheumatoid Arthritis. Journal of inflammation research, 15, 1173–1181. https://doi.org/10.2147/JIR.S350281
Wan, Y., & Fu, J. (2024). GDF15 as a key disease target and biomarker: linking chronic lung diseases and ageing. Molecular and cellular biochemistry, 479(3), 453–466. https://doi.org/10.1007/s11010-023-04743-x
Comment 7: Please explain the clinical significance that GDF-15 was significantly higher in the acute phase than in the intermission phase, but the ROC cut-off value was close (1160.8 vs. 1147.5 pg/mL).
Response 7: We thank the reviewer for this insightful comment. While serum GDF-15 levels were significantly higher during the acute phase compared to the intercritical phase, the ROC cut-off values for differentiating patients from healthy controls were relatively close. This finding can be explained by the observation that GDF-15 levels were already markedly elevated in intercritical gout patients compared to healthy individuals, resulting in a high cut-off threshold even outside the acute flare. In the acute phase, an additional increase in GDF-15 levels occurs. Still, since the ROC analysis aims to optimize sensitivity and specificity in distinguishing any disease state from the healthy group, both phases yield similarly high cut-off values. This supports the notion that GDF-15 is consistently elevated in gouty arthritis and may serve as a robust biomarker across disease phases, rather than only during acute inflammation.
Osman CURE (M.D.)
On behalf of the Authors
Round 2
Reviewer 2 Report
Comments and Suggestions for Authors
The author has revised the article in accordance with the requirements. I have no further questions about this MS.